# Mycn regulates intestinal development through ribosomal biogenesis in a zebrafish model of Feingold syndrome 1

Yun-Fei Li[1☯‡], Tao Cheng[1,2☯‡], Ying-Jie Zhang[1], Xin-Xin Fu[1], Jing Mo[3], Guo-Qin Zhao[3], Mao-Guang Xue[1], Ding-Hao Zhuo[1,2], Yan-Yi Xing[1,2], Ying Huang[1], Xiao-Zhi Sun[1], Dan Wang[1], Xiang Liu[1], Yang Dong[1,2], Xiao-Sheng Zhu[1], Feng He[1,2], Jun Ma[1,2], Dong Chen[4], Xi Jin[5]*, Peng-Fei Xu[1,2]*

1 Institute of Genetics and Department of Human Genetics, Zhejiang University School of Medicine, Hangzhou, China, 2 Women's Hospital, Zhejiang University School of Medicine, Hangzhou, China, 3 Department of Immunology, Guizhou Medical University, Guiyang, China, 4 Department of Colorectal Surgery, The First Affiliated Hospital of Zhejiang University School of Medicine, Hangzhou, China, 5 Department of Gastroenterology, The First Affiliated Hospital of Zhejiang University School of Medicine, Hangzhou, China

☯ These authors contributed equally to this work.
‡ These authors share first authorship on this work.
* jxfl007@zju.edu.cn (XJ); pengfei_xu@zju.edu.cn (P-FX)

**Data Availability Statement:** All bulk RNA-seq, single cell RNA-seq data and Ribo-seq files are available from the GEO database (accession number: GSE191003, GSE211652).

## Abstract

Feingold syndrome type 1, caused by loss-of-function of MYCN, is characterized by varied phenotypes including esophageal and duodenal atresia. However, no adequate model exists for studying the syndrome's pathological or molecular mechanisms, nor is there a treatment strategy. Here, we developed a zebrafish Feingold syndrome type 1 model with nonfunctional *mycn*, which had severe intestinal atresia. Single-cell RNA-seq identified a subcluster of intestinal cells that were highly sensitive to Mycn, and impaired cell proliferation decreased the overall number of intestinal cells in the *mycn* mutant fish. Bulk RNA-seq and metabolomic analysis showed that expression of ribosomal genes was down-regulated and that amino acid metabolism was abnormal. Northern blot and ribosomal profiling analysis showed abnormal rRNA processing and decreases in free 40S, 60S, and 80S ribosome particles, which led to impaired translation in the mutant. Besides, both Ribo-seq and western blot analysis showed that mTOR pathway was impaired in *mycn* mutant, and blocking mTOR pathway by rapamycin treatment can mimic the intestinal defect, and both L-leucine and Rheb, which can elevate translation via activating TOR pathway, could rescue the intestinal phenotype of *mycn* mutant. In summary, by this zebrafish Feingold syndrome type 1 model, we found that disturbance of ribosomal biogenesis and blockage of protein synthesis during development are primary causes of the intestinal defect in Feingold syndrome type 1. Importantly, our work suggests that leucine supplementation may be a feasible and easy treatment option for this disease.

**Funding:** The authors acknowledge support by grants the Chinese National Key Research and Development Project (2019YFA0802402, 2018YFC1003203) and the National Scientific Foundation of China (32050109, 31970757, 31900576) to PX. The funders had no role in study design, data collection and analysis, decision to publish or preparation of the manuscript.

**Competing interests:** The authors have declared that no competing interests exist.

**Abbreviations:** DCFH-DA, 2′,7′-Dichlorodihydrofluorescein diacetate; DIG, digoxigenin; dpf, days postfertilization; ESC, embryonic stem cell; FPKM, fragments per kilobase of exon per million mapped fragments; GO, gene ontology; GSEA, gene set enrichment analysis; HE, hematoxylin–eosin; hpf, hours postfertilization; ifabp, intestine fatty acid–binding protein; ISH, *in situ* hybridization; PBS, phosphate-buffered saline; PFA, paraformaldehyde; S6K1, S6 kinase 1; TOR, Target of rapamycin; WISH, whole mount *in situ* hybridization; WT, wild-type; 4EBP1, 4E-binding protein 1.

## Introduction

Feingold syndrome is a skeletal dysplasia caused by loss-of-function mutations of either MYCN (type 1) or MIR17HG (type 2), which encodes miR-17-92 microRNAs [1]. The syndrome is characterized by autosomal dominant inheritance of microcephaly and limb malformations, notably hypoplastic thumbs, and clinodactyly of the second and fifth fingers. Feingold syndrome type 1 is always accompanied by gastrointestinal atresia (primarily esophageal and/or duodenal atresia) [2]. However, whether the digestive system deficiency in Feingold syndrome 1 is the direct result of MYCN loss-of-function or a sequence effect of other developmental defects caused by MYCN mutation remains unknown. Further, the mechanism of the intestinal defects in patients with Feingold syndrome type 1 is also unclear. This unclear pathogenesis of Feingold syndrome type 1 is a major obstacle to developing treatments for the disease.

The MYC proto-oncogene family is a class of transcription factors with a basic helix–loop–helix domain and includes MYC, MYCL, and MYCN [3]. MYCN amplification or overexpression has been described in many cancers, including neuroblastoma, retinoblastoma, rhabdomyosarcoma, and lung cancer, which are frequently of embryonic or neuroendocrine origin [4]. Like other members of the MYC family, MYCN controls the expression of its target genes and regulates many fundamental cellular processes such as proliferation, differentiation, apoptosis, protein synthesis, and metabolism [5]. Research on chicken embryos has shown that overexpression of MYCN drives the neural crest toward a neural stem cell fate [6]. MYCN homozygous mutant mice die at embryonic day E11.5, and multiple organs, including the nervous system, mesonephros, lungs, and gut, are affected. Besides, this study show that the homozygous mutant embryos bleed easily upon manipulation, exhibit distended aortas, and are severely anemic when examined at E12. These observations suggest a failure in the developing cardiovascular system, leading to spontaneous bleeding that would also result in embryonic death [7]. Conditional disruption of MYCN in mouse neural progenitor cells has shown that MYCN is essential for neural progenitor cell expansion and inhibits its differentiation [8]. However, the function of MYCN in organogenesis remains uncertain, and the mechanism through which MYCN regulates intestinal development remains unknown.

The vertebrate alimentary canal is derived from the primitive gut tube, which originates from the endodermal layer [9] and gives rise to the digestive system organs, including the pancreas, liver, gall bladder, and intestines. Developmental defects in this process can lead to serious human diseases, such as intestinal atresia, malrotation, hypoplasia, and epithelial defects, which cause malabsorptive or secretory diarrheal syndromes [10]. Although the zebrafish digestive system differs morphologically from that of mammals, a high degree of homology exists between zebrafish intestines and mammalian intestines in terms of their cellular composition and molecular pathways regulating intestinal development [11]. Many zebrafish models for studying congenital diseases affecting the intestines have been reported because of experimental tractability; these models have provided novel insights into the developmental mechanisms, pathogenesis, and therapeutics of intestinal congenital diseases [12,13].

In this study, we generated a large deletion in *mycn* in zebrafish using the CRISPR/Cas9 system. Homozygous *mycn* mutant fish are viable and fertile, and most importantly, the mutants carry a series of developmental defects similar to those of Feingold syndrome type 1, such as an abnormal pharyngeal arch (cartilage defects) and intestinal deficiency. Using this model, we studied the mechanism of the Mycn deficiency leading to intestinal developmental defects and discovered a potential therapeutic strategy for alleviating the intestinal defects in patients with Feingold syndrome type 1.

## Material and methods

### Ethics statement

All animal procedures were performed per the requirements of the "Regulation for the Use of Experimental Animals in Zhejiang Province." The Animal Ethics Committee of the School of Medicine, Zhejiang University, approved this study. The protocol number is NO. 24278.

### Fish lines and maintenance

The zebrafish AB strain was used in all experiments to generate knock-in or mutant lines. To generate the *mycn* mutants, we synthesized 3 gRNAs against the second exon of the zebrafish *mycn* gene as previously described [14]. The Cas9 protein and *mycn*-targeting gRNAs were coinjected into the wild-type (WT) embryos at 1-cell stage. The *mycn* mutant lines were identified in the F1 generation by analyzing the PCR product using the primer pair listed in S1 Table. To construct *mycn*:*EGFP* knock-in zebrafish, we generated a gRNA targeting the second intron, and a donor DNA with EGFP reporter just before the stop codon of *mycn* flanked with 2 homologous arms. We coinjected Cas9, gRNA, and the donor into zebrafish embryos at the 1-cell stage, then screen embryos with correct EGFP expression.

### WISH

Embryos were fixed in 4% paraformaldehyde (PFA) in phosphate-buffered saline (PBS) overnight at 4˚C. The probes were labeled with digoxigenin (Roche Diagnostics). The *ifabp*, *lfabp*, *prss1*, *insulin*, *foxa3*, *foxa1*, *gata6*, *slc15a1b*, *pyyb*, *rpls*, *rps*, *lmo7a*, *baiap2l1a*, *fabp6*, *id2a*, *MT-ND2*, *tcnl*, *tm4sf4*, and *apoc2* probes were generated and used for whole mount *in situ* hybridization (WISH) as previously described [15]. DNA fragments of above genes were cloned into the pEASY-blunt-zero vector (Transgen), then sequenced. Primers used were listed in S1 Table.

### Paraffin sections, hematoxylin–eosin (HE) staining, immunofluorescence staining, and TUNEL assay

Embryos were fixed in 4% PFA overnight at 4˚C for immunofluorescence or HE staining. After washing in PBST (0.1% Tween20 in PBS), the embryos were dehydrated, transparentized with xylene, and mounted in paraffin overnight at 4˚C. The sections were cut serially to 3-μm thick and collected on poly-L-lysine-coated glass slides (CITOGLAS, 188105). HE staining was performed using Dyeing and sealing machine (GEMINI AS). Immunofluorescence staining was performed as described [16]. PCNA antibody was purchased from Sigma (P8825, 1:1,000); puromycin antibody was purchased from Merck (MABE343, 1:1,000). Alexa Fluor 488–labeled secondary antibody (Invitrogen A21206) was used for visualization. The TUNEL assay was performed with the *In Situ* Cell Death Detection Kit (Roche) per the manufacturer's instructions.

### Alcian blue staining and alizarin red staining

The 5 days postfertilization (dpf) embryos were fixed in 4% PFA at 4˚C overnight. The embryos were washed several times with PBS, then stained with 0.1% Alcian blue 8GX in acid alcohol (70% ethanol and 30% glacial acetic acid) overnight at room temperature. After rehydration in PBS, the embryos were digested with 1% trypsin for 1 h at 37˚C. Embryos were then washed with PBS several times and stored in glycerol.

The 7 dpf embryos were fixed in 4% PFA with 1% 5N NaOH at 4°C overnight. The embryos were washed several times with PBS, then stained with Alizarin red solution (0.4 ml saturated Alizarin red S in ethanol/10 ml 0.5% KOH) overnight at room temperature. After wash in 0.5% aqueous KOH for several times, the embryos were transferred in graded series of glycerol (15%, 30%, 60%, and 80% in 0.5% KOH) and stored in 100% glycerol.

## Western blot

Proteins of 3 dpf *mycn* mutant and WT zebrafish were harvested in 100 μL RIPA buffer separately. The protein lysates were separated via SDS-PAGE and transferred to a polyvinylidene difluoride membrane (Millipore). The western blot was performed using anti-puromycin (1:1,000, Merck), anti-mouse-horseradish peroxidase (1:5,000, Thermo Fisher, UK), anti-tubulin (1:10,000, Sigma), anti-RPS6 (1:1,000, ABclonal), anti-Phospho-RPS6 (1:1,000, ABclonal), anti-4EBP1 (1:1,000, 9644T, CST), and anti-Phospho-4EBP1 (1:1,000, 2855T, CST).

## Sucrose-gradient centrifugations

The sucrose-gradient centrifugations were performed as previously described [17]. WT and *mycn* mutant embryos were collected at 3 dpf, digested with cold 0.5% trypsin to a single-cell suspension, and homogenized in lysis buffer for 20 to 50 times. The homogenate was centrifuged at 12,000*g* for 15 min at 4°C, and the supernatants were layered on top of a 10% to 50% sucrose-gradient solution. Ultracentrifugation was performed at 36,000 rpm for 2 h (hours) at 4°C. After centrifugation, the absorbance at optical density OD260 of the fractions collected from the top of the tube was detected using a TRAIX detector (Thermo Scientific, USA).

## Chemical treatment

2′,7′-Dichlorodihydrofluorescein diacetate (DCFH-DA) (5 mg/mL, Sigma) was used to label the zebrafish intestines at 7 dpf. Embryos were treated with DCFH-DA in 0.3× Danieau buffer for 2 h. PED6 (D23739, Thermo Fisher) and EnzChek (E6639, Thermo Fisher) were used to test the digestive ability of the intestinal proteins and lipids. Embryos at 7 dpf were treated with 3 μg/mL PED6 or 20 μg/mL EnzChek in a 0.3× Danieau buffer for 3 h. Rapamycin was used to inhibit the mTORC1 pathway. Embryos were exposed to 400 and 800 nM rapamycin (Sangon Biotech, China) in a 0.3× Danieau buffer from 10 hours postfertilization (hpf) to 3 dpf. L-Leu and *rheb* mRNA were used to elevate the mTORC1 pathway. Embryos were injected with L-Leu (500 nM, Sigma) at 30 hpf or *rheb* mRNA (100 pg and 150 pg) at 1-cell stage, then harvested at 3 dpf.

## Flow cytometry assay

WT and *mycn* mutant embryos with *ET33J1*: *EGFP* transgenic backgrounds were collected at 3 dpf. For proliferation analysis, the embryos were immersed in 3 mg/mL BrdU in 0.3× Danieau buffer at 60 hpf overnight in the dark, then washed 3 times with PBST and digested with cold 1% trypsin to a single-cell suspension. The cells were incubated and labeled with BrdU primary antibody and Alexa Fluor 594–labeled secondary antibody. For apoptosis analysis, the cells were stained with the Annexin V-APC/7-AAD apoptosis kit (Multisciences) per the manufacturer's instructions, and the signal was detected by a DxFLEX flow cytometer.

## Northern blot

Total RNA was extracted from fish samples using TRIpure Reagent (Aidlab, RN0102) according to the manufacturer's instructions. Digoxigenin (DIG)-labeled 5′ETS-1, ITS1, and ITS2

probes were obtained by PCR with specific primers (S1 Table) and the corresponding plasmid DNA as the template, together with the DIG DNA Labeling Mix (Roche Diagnostics,11277065910). Northern blot hybridization was performed as previously described [18]. DIG-labeled probes were detected with CDP-Star Chemiluminescent Substrate (Roche, Cat#12041677001), according to the manufacturer's instructions.

## Ribo-seq

WT and *mycn* mutant embryos were collected at 3 dpf, digested with cold 0.5% trypsin to a single-cell suspension, and homogenized in lysis buffer for 20 to 50 times. The homogenate centrifuged at 12,000*g* for 15 min at 4˚C, and the supernatants were digested with RNase for 30 min and aborted by 1 M EGTA followed by layering on top of a 10% to 50% sucrose-gradient solution. Ultracentrifugation was performed at 36,000 rpm for 2 h at 4˚C. After centrifugation, the fractions were collected according to the absorbance at optical density OD260 by a TRAIX detector. Then the RNA was purified and used to construct the library for sequencing.

## Processing of ribosome profiling data

FastQC (https://github.com/s-andrews/FastQC) was used to perform quality issues inspection for raw fastq files. For quality trimming, cutadapt (http://cutadapt.readthedocs.org/en/stable/) was used to remove adapter sequences and filter out reads that became shorter than 20 nt (-m parameter). Trimmed clean reads were aligned to the zebrafish GRCz11 genome using STAR (https://github.com/alexdobin/STAR), and reads mapped to multiple genomic location were removed. featureCounts [20] was used to calculated gene expression counts of each sample. The expression counts of all samples were transformed to fragments per kilobase of exon per million mapped fragments (FPKM). Translation efficiency was defined by the ratio of Ribo-seq FPKM and RNA-seq FPKM. Differential translation efficiency analysis was performed by R package limma (https://bioconductor.org/packages/limma/). More than 3,000 genes were identified as down-regulated genes in *mycn* mutant samples ($p$-value $< 0.05$ and log2 fold change $< -0.5$). Those genes were uploaded to KOBAS (http://bioinfo.org/kobas/) web tool to perform KEGG pathway enrichment analysis ($p$-value $< 0.01$, Edge weight = 0.2, Top cluster = 3).

## Processing of GSEA

For gene set enrichment analysis (GSEA), the mappings linking zebrafish gene to gene ontology (GO) terms were achieved through org.Dr.eg (https://bioconductor.org/packages/org.Dr.eg.db/). A ranked list was formed for GSEA using sign (log2FC) ∗(−log10PValue) as the ranking statistic. GSEA was performed by clusterProfiler package (https://bioconductor.org/packages/clusterProfiler/), using fgsea (https://github.com/ctlab/fgsea) with 10e5 interactions, and the results were visualized by enrichplot package (https://bioconductor.org/packages/enrichplot/).

## RNA-seq and GO analysis

WT and *mycn* mutant zebrafish embryos were collected for RNA sequencing at 2 and 3 dpf. Library construction and sequencing were completed by Novogene (Novogene Bioinformatics Technology, Beijing, China). Paired-end sequencing (Novaseq6000, 150-bp reads) was performed. The sequencing reads were aligned to the zebrafish GRCz11 genome using STAR (https://github.com/alexdobin/STAR) [19], and reads mapped to multiple genomic locations were removed. Gene expression counts for each sample were calculated using featureCounts

[20]. Differential expression analysis was performed using the DESeq2 package (https://github.com/mikelove/DESeq2) [21]. Differentially expressed genes were obtained by comparing the *mycn* mutant to the WT samples with padj $\leq$ 0.1 and |log2foldchange| < 0. Finally, overlapped down-regulated genes of the 2 and 3 dpf samples were selected. GO biological process analysis was performed with the clusterProfiler package (https://bioconductor.org/packages/clusterProfiler/) [22].

## Single-cell RNA-seq library preparation and sequencing

Samples were prepared for the single-cell RNA-seq as previously described [23]. Approximately 30 *mycn* mutant or WT zebrafish embryos at 3 dpf were transferred to 1.5 mL low binding microcentrifuge tubes (Eppendorf 022431021). Trypsin-EDTA solution (Beyotime, C0201) was added, then the solution was pipetted up and down several times through a P200 tip every 5 min for 30 min. After all embryos were dissociated into single cell, the cells were centrifuged into a pellet and resuspended by adding 200 μL 0.05% bovine serum albumin/Ringer's solution. Cell density was quantified manually using hemacytometers (QIUJING), and cell viability was analyzed using propidium iodide staining. Libraries were prepared using the Chromium Controller and Chromium Single Cell 3′ Library (10× Genomics, PN-1000074) per the manufacturer's protocol for 10,000-cell recovery. Final libraries were sequenced on the Illumina Novaseq6000 (Genergy Bio-technology, Shanghai).

## Single-cell RNA-seq analysis

Raw sequencing reads were analyzed using the 10X Cellranger pipeline, version 3.0.2, with the default parameters. The expression matrix was obtained after running Cellranger. The R package Seurat (version 4.0.2) [24] was used to perform downstream analysis. We created a Seurat object with the CreateSeuratObject() function with min.cells = 3, min.genes = 100. Next, we performed a standard analysis procedure with the functions FilterCells(), NormalizeData(), FindVariableGenes(), FindClusters(), and FindAllMarkers(), with appropriate parameters. Finally, all clusters were visualized in 2 dimensions using t-SNE or UMAP. These clusters were annotated based on differentially expressed markers in each cluster or by comparison with published single-cell datasets. The WT and *mycn* mutant datasets were integrated using the Seurat integration procedure. First, variable features for each dataset were normalized and identified independently with nfeatures = 2,000. The FindIntegrationAnchors function was used to identify anchors; the anchors were used as input for the IntegrateData function to integrate the 2 datasets. Finally, the same procedures were performed on the integrated datasets as done for the single dataset with appropriate parameters. The Seurat subset function was used to create an intestinal cell Seurat object for downstream analysis.

## Results

### *mycn* was highly enriched in the developing digestive system of the zebrafish embryos

To use zebrafish as a model for studying Feingold syndrome type 1 and the function of Mycn during organogenesis, we first explored the spatiotemporal expression patterns of *mycn* during embryonic development in zebrafish. *mycn* transcription can be detected by RNA *in situ* hybridization (ISH) at the onset of gastrulation and is enriched in the neural ectoderm. At the end of gastrulation, *mycn* was specifically expressed in both the anterior and posterior neural plate, consistent with previous reports on the role of MYCN in neural development and oncogenesis [25]. Beginning at 18 hpf, *mycn* expression could be detected in the epiphysis, eye,

optic tectum, spinal cord, and endoderm (S1A Fig). After 24 hpf, *mycn* expression was progressively restricted to the central nervous system, pharyngeal arch, and digestive system (Fig 1A).

To validate the ISH results and investigate the *mycn* expression dynamics during zebrafish development, we generated an *mycn*:*EGFP* knock-in fish by inserting an EGFP sequence just before the stop codon of *mycn* with a P2A linker between the 2 proteins (S1C Fig) to avoid possibly distorting the Mycn protein structure. The *mycn*:*EGFP* line confirmed that *mycn* was mainly expressed in the central nervous and digestive systems during organogenesis (Fig 1C). Interestingly, *mycn* was also expressed in the migrating neuromast cells of the lateral line, indicating that Mycn might also function in the sensory organs. High-magnification imaging of both the *mycn* ISH and *mycn*:*EGFP* showed that *mycn* was expressed in the intestinal epithelial cells, suggesting that Mycn functions directly in intestinal development (Fig 1B and 1D).

To further characterize the *mycn* expression pattern, we performed single-cell RNA-seq for WT embryos at 3 dpf. A total of 23 clusters were identified and annotated after strict quality control (Fig 1E). We then explored the *mycn* expression level across these clusters and found that *mycn* was highly expressed in the central nervous system, neural crest cells, and endoderm-derived tissues such as the intestines, liver, and pancreas (Fig 1F). Additionally, we investigated *mycn* expression in published single-cell RNA-seq datasets [26–28]. During the gastrulation and somitogenesis stages, the brain and optic cells showed high *mycn* expression (S1D and S1E Fig). High *mycn* expression levels were detected in intestinal cells at 2 dpf, and *mycn* expression decreased in the intestines at 5 dpf (S1B and S1F–S1I Fig), which was consistent with the ISH experiment on *mycn*.

## *mycn* mutation results to multiple developmental defects in zebrafish recapitulating human Feingold syndrome type1

To construct a zebrafish model of Feingold syndrome and study the function of Mycn during development, we generated a *mycn* mutant fish line using the CRISPR/Cas9 system. Three gRNAs targeting exon2 of *mycn* were injected together with Cas9 into zebrafish embryos at the 1-cell stage, resulting in a 308-bp deletion in the CDS (coding sequence) of *mycn* (S2A Fig). This large deletion was easily discriminated via PCR and led to early termination of the Mycn protein (122 amino acids left) (S2B Fig). Homozygous Mycn mutation leads to embryonic lethality in mammalian system; we also found that most of our *mycn* mutant zebrafish died at around 10 dpf (S2C Fig). However, some of the homozygotic *mycn* mutant fish survived to adulthood and were fertile.

The WT and *mycn* mutant zebrafish did not noticeably differ morphologically until 4 dpf; at this stage, the *mycn* mutants lacked swim bladder, and their heads and eyes were smaller than those of the WT fish (S2E Fig), but those defects become less obvious after 1 day of development in some of the mutants (Fig 2A). We measured the body length at 3 dpf and 4 dpf in WT and *mycn* mutant and found there is no significant difference (S2D Fig), suggesting that the above phenotypes were not result from whole developmental retardation. Alcian blue staining showed that the pharyngeal arch was severely malformed in most of the 5 dpf *mycn* mutants (Fig 2B). However, the skull development was normal in the *mycn* mutant, showed by alizarin red staining at 7 dpf (S2F Fig). Interestingly, ISH showed decreased expression of *shha* and *hand2* in the fin bud of *mycn* mutants (Fig 2E), which may mimic the abnormal hands development in Feingold syndrome [29].

Gastrointestinal atresia is another of the most distinctive yet least studied symptoms of Feingold syndrome type 1. The highly enriched expression of *mycn* mRNA in the developing intestines of zebrafish suggests that Mycn has a direct function in intestinal development. To

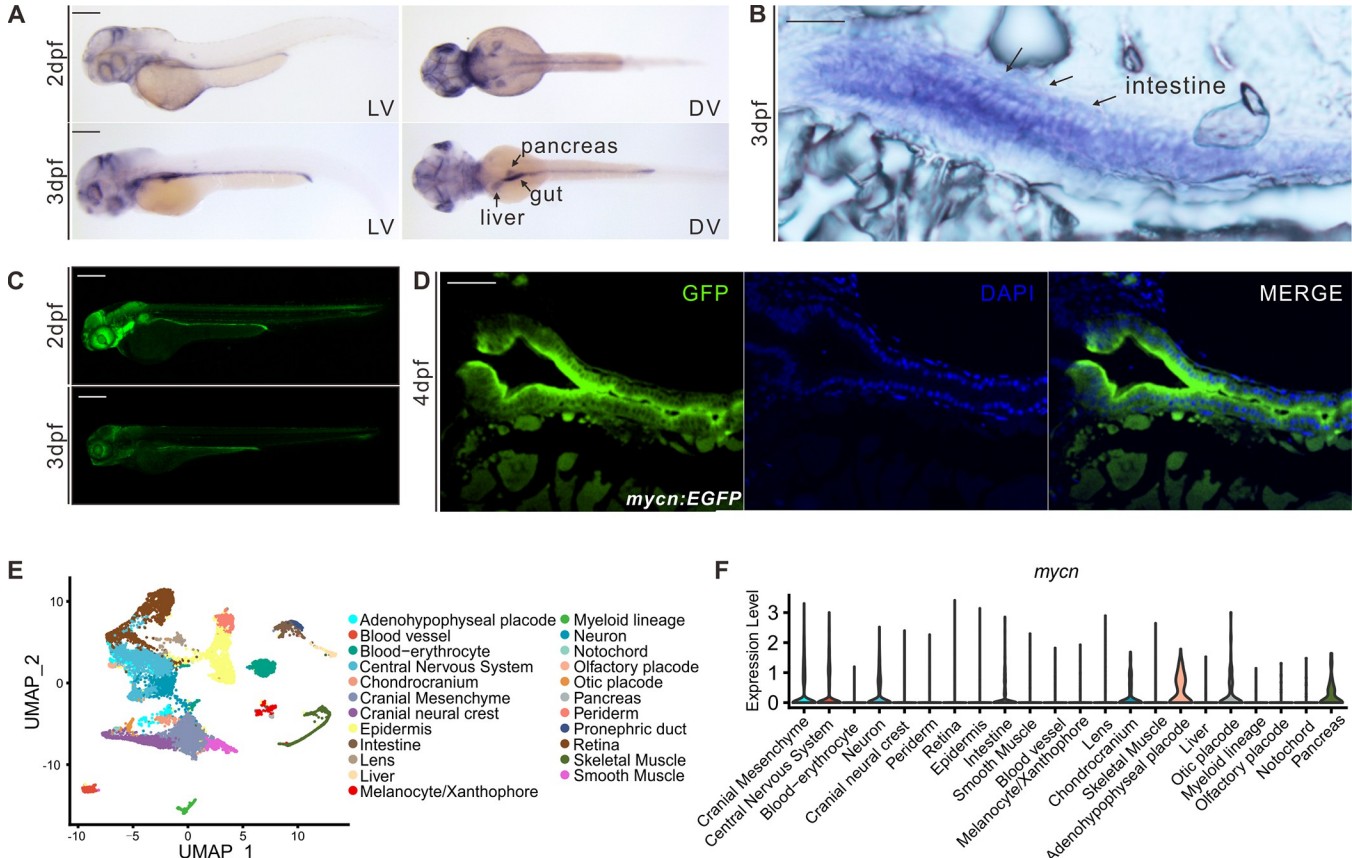

**Fig 1. *mycn* expression patterns in zebrafish during early development.** (A) Expression patterns of *mycn* in zebrafish at 2 and 3 dpf by whole-mount WISH. Lateral view (left), dorsal view (right). (B) *mycn* expressed along the whole intestines of the embryos at 3 dpf shown by section via ISH. Arrows indicate the embryo intestines. Sections were cut along the sagittal plane. (C) Fluorescence images show the *mycn* expression patterns by EGFP-knock-in fish at 2 and 3 dpf (lateral view). (D) *mycn* expressed along the whole intestines of the zebrafish at 4 dpf shown by section of *mycn*:*EGFP* fish. (E) UMAP plot shows unsupervised clustering of the cells in WT embryos of 3 dpf; cells are colored by their cell type annotations. (F) Violin plots show the *mycn* expression levels of different cell types of WT embryonic scRNA-seq data at 3 dpf. Scale bars: 200 μm (A and C), 50 μm (B and D). dpf, days postfertilization; ISH, *in situ* hybridization; scRNA-seq, singe-cell RNA-seq; UMAP, uniform manifold approximation and projection; WISH, whole mount *in situ* hybridization; WT, wild-type.

investigate this, we firstly crossed our *mycn* mutant fish with a gut epithelial reporter *ET33J1*: *EGFP* zebrafish line [30]. The reporter showed that the size of the entire intestine of the *mycn* mutants at 5 dpf was dramatically smaller than that of the WT fish (Fig 2C). DCFH-DA treatment at 7 dpf also confirmed the intestinal developmental defect in the *mycn* mutants (Fig 2D). Besides, the length of intestine in *mycn* mutant is also significantly shorter compared with WT (Fig 2D). HE staining of the intestinal sagittal section showed that in the anterior region of the *mycn* mutant intestine (intestinal bulb), the intestinal folds were obviously smaller and fewer, the mid and posterior intestines of the *mycn* mutants showed an obviously narrowed lumen, and the goblet cells, which emerge in the middle of the intestines, could hardly be detected. Conversely, the goblet cells in the WT fish could be easily detected at 4 dpf (Fig 2F). WISH results also showed that expression of the intestinal marker, intestine fatty acid–binding protein (*ifabp*), was drastically reduced, likely due to the reduced intestinal size (Fig 2G).

To verify that *mycn* loss-of-function, rather than off-target effects, caused these phenotypes, we performed a rescue experiment in the *mycn* mutant embryos by injecting a plasmid with the full-length *mycn* and an EGFP sequence under the *sox17* promoter. The *sox17-mycn* cassette was flanked by tol2 transposons, and transposase mRNA was coinjected to improve the expression efficiency (Fig 2H). The intestinal size was significantly restored in the embryos

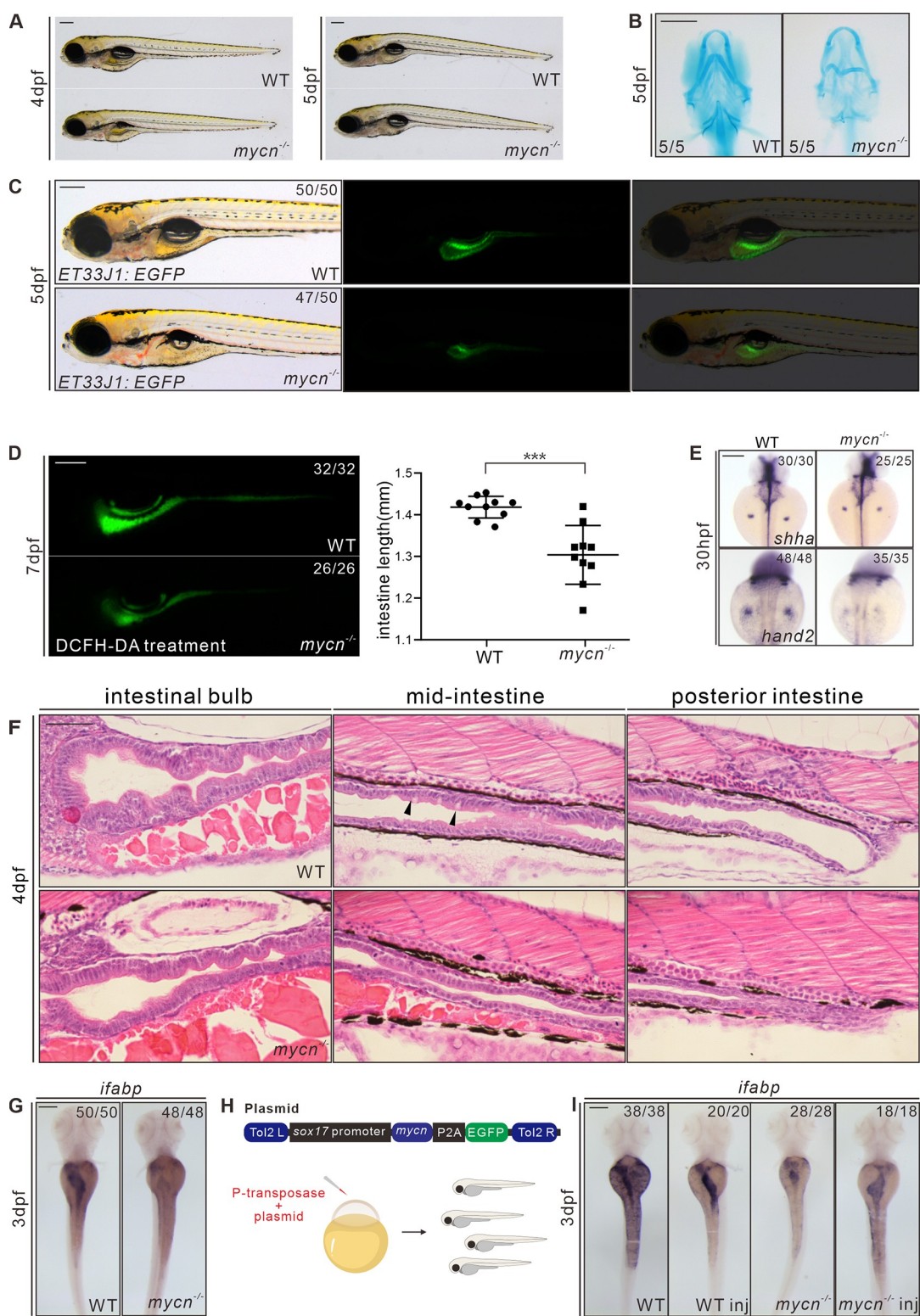

**Fig 2. Phenotypic analysis of *mycn* mutants. (A)** Bright-field images show that the *mycn* mutants lacked swim bladder at 4 dpf but appeared at 5 dpf. All embryos are shown in lateral view. **(B)** Abnormal pharyngeal arch development was observed in *mycn* mutants via Alcian blue staining; images show the head region of WT and *mycn* mutant embryos at 5 dpf (ventral view). **(C)** Intestinal lumens in WT and *mycn* mutant embryos at 5 dpf shown by *ET33J1: EGFP* reporter (lateral view). **(D)** Fluorescence signals show the morphology of the whole intestines in WT and *mycn* mutant embryos at 7 dpf via DCFH-DA

treatment (lateral view); statistical analysis of intestine length was showed in right. **(E)** Expression of *shha* and *hand2* by WISH showing fin bud development in WT and *mycn* mutant embryos at 30 hpf (dorsal view). **(F)** Morphology of the whole intestine visualized via HE staining for the WT and *mycn* mutant embryo sections at 4 dpf. Black arrows indicate goblet cells in the WT intestines. Sections were cut along the sagittal plane. **(G)** Expression of *ifabp* by WISH showing intestines in WT and *mycn* mutant embryos at 3 dpf (dorsal view). **(H)** Diagram of the rescue plasmid construction. Overexpression of *mycn* specifically in endoderm cells was driven by *sox17* promoter. **(I)** Expression of *ifabp* in WT (2 images on the left) and *mycn* mutant embryos (2 images on the right) at 3 dpf. The injected embryos are labeled with "inj" at the bottom of the image. Scale bars: 200 μm (**A**-**F**), 50 μm (**G**). The data underlying this figure can be found in S1 Data. DCFH-DA, 2′,7′-Dichlorodihydrofluorescein diacetate; dpf, days postfertilization; HE, hematoxylin–eosin; hpf, hours postfertilization; *ifabp*, intestine fatty acid–binding protein; WISH, whole mount *in situ* hybridization; WT, wild-type.

that had strong endodermal EGFP expression, as shown by the *ifabp* expression (Fig 2I). These results confirmed that the intestinal defects in the mutant fish resulted from the loss-of-function of Mycn and suggest that Mycn functions directly in endoderm development.

In addition to the intestines, the primitive gut tube (foregut) gives rise to the liver and pancreas. Thus, we performed WISH using markers of the liver (*lfabp*), the exocrine pancreas (*prss1*), and islets (*insulin*) in the WT and *mycn* mutant embryos at 3 dpf. The *mycn* mutants displayed a smaller liver and exocrine pancreas, while islet development appeared unaffected (S3A Fig). To trace the earliest discernable stage of the endoderm phenotypes in the *mycn* mutants, we performed WISH using the pan-endodermal markers *foxa1*, *foxa3*, and *gata6* in embryos at 30 and 48 hpf. At 30 hpf, when the liver and pancreas buds began to emerge, these markers did not obviously differ between the WT and *mycn* mutant embryos. This suggests that Mycn loss-of-function did not strongly affect differentiation of the liver and pancreas during their budding stages. However, the smaller liver, pancreas, and intestines at 48 hpf indicated that Mycn plays an important role during outgrowth of these digestive organs (S3B Fig).

To further analyze the functional defects in the digestive tracts of the *mycn* mutants, we treated the embryos with PED6 (for lipid-processing activity) or EnzChek (for protein-processing activity) (S4A Fig) [31]. Although these activities were severely compromised overall, the fluorescence intensity appeared normal in the mutants, suggesting that although the mutant intestinal morphology was highly defective, its main absorption and digestive functions were not completely lost. Thus, we further analyzed the 3 main functional cells of the intestines: the absorptive, enteroendocrine, and goblet cells [32]. Periodic acid–Schiff and Alcian blue staining of the sagittal sections indicated that goblet cells was dramatically reduced but could still be observed in the *mycn* mutants at 5 dpf (S4B Fig). HE staining also confirmed this (Fig 2F). For the enteroendocrine and absorptive cells, we performed WISH using *pyyb* (enteroendocrine cell marker) and *slc15a1b* (absorptive cell marker) probes. Although both could be detected, expression of these markers was dramatically decreased in the *mycn* mutants (S4C Fig). In summary, Mycn loss-of-function led to abnormal development of the entire digestive tract. Although differentiation of the main functional cells was unaffected, Mycn loss-of-function caused a decrease in the overall intestinal size and further weakened the digestive system functions.

In addition to the endodermal defects, the enteric neurons were significantly reduced in the mutants compared with those of the WT fish at 5 dpf, in terms of both fluorescence intensity and cell numbers, as demonstrated by the anti-HuC antibody (S4D Fig). Whether this phenotype was due to the direct loss-of-function of Mycn in the enteric neuron or was a sequence effect of the intestinal atresia should be further investigated.

## Single-cell transcriptomics showed that Mycn loss-of-function reduced the specific intestinal cell types during embryonic development

To systematically investigate the phenotypes resulting from Mycn loss-of-function at a higher resolution, we performed single-cell RNA-seq in the *mycn* mutant fish at 3 dpf. We integrated

*mycn* mutant scRNA-seq datasets with WT datasets (see Material and methods). A total of 27 cell types were identified and annotated based on their expression markers (Figs 3A and S5A). Calculating the cell ratio of each cell type revealed that the cranial neural crest, blood vessels, and digestive organs, including the intestines, liver, and pancreas, were dramatically decreased in the *mycn* mutants (Fig 3B). To more deeply characterize the intestinal cell clusters, we selected and reclustered the intestinal cells and identified 9 subclusters (Fig 3C). We noticed that cluster 8 highly expressed *mycn*, and this cluster showed stemness characteristics based on the pseudotime trajectory analysis (S5C and S5D Fig); this cluster was completely disappeared in the *mycn* mutants. Besides, clusters 4 and 6 were significantly reduced in the *mycn* mutant. To characterize these subclusters, we performed differential expression analysis, selected the most significant markers (Fig 3D), and verified their expressions via ISH in both WT and *mycn* mutant embryos. Expressions of all those selected cluster markers were decreased in the *mycn* mutant intestines at 3 dpf, especially for those of clusters 4, 6, 7, and 8, which could hardly be detected in the mutants (Fig 3E).

## Proliferation arrest, but not apoptosis, led to intestinal defects in the *mycn* mutants

The morphological, molecular, and single-cell RNA-seq results indicated that the intestinal defects in the *mycn* mutants may be due to reduced cell numbers rather than to a differentiation blockade. Thus, we analyzed the levels of proliferation and apoptosis in the intestines of the *mycn* mutant and WT embryos at 3 dpf. Immunofluorescence showed that the PCNA signals were significantly reduced in the intestinal bulbs of the *mycn* mutants (Fig 4A). However, TUNEL assay revealed no obvious apoptotic signals in the intestinal bulb region of either the *mycn* mutant or WT embryos (Fig 4B). To further confirm the intestinal proliferation defects of the *mycn* mutants, we dissociated the 3 dpf embryos of the *mycn*$^{-/-}$; *ET33J1*: *EGFP* line and performed fluorescence-activated cell sorting to sort out the intestinal cells. We then used BrdU and Annexin V-APC analysis to mark the cells that were proliferating or undergoing apoptosis, respectively. Consistent with the immunofluorescence results, the flow cytometry results also showed a significantly lower proliferation rate in the *mycn* mutants, but apoptosis could hardly be detected in the intestines of either the WT or mutant embryos at 3 dpf (Fig 4C–4E).

## Protein translation and nucleotide biosynthesis were compromised in the *mycn* mutants

To understand the molecular mechanism of the decreased proliferation that led to intestinal developmental defects in the *mycn* mutants, we performed bulk RNA-seq in the *mycn* mutant and WT embryos at 2 and 3 dpf. Compared with the WT fish, we identified 470 down-regulated and 287 up-regulated genes in 2 dpf *mycn* mutants and 451 down-regulated genes and 409 up-regulated genes in 3 dpf *mycn* mutants, respectively (Figs 5A and S6A and S6B and S3 and S4 Tables). Among them, 85 genes were down-regulated in both stages (Fig 5A). GO enrichment analysis revealed that these 85 genes were mainly involved in the processes of ribosomal assembly, translation, nucleotide/nucleoside biosynthesis, and nucleoside metabolism (Fig 5B and 5D). Interestingly, interaction networks of enriched GO terms revealed that the digestive system development, including that of the pancreas, was closely related to the ribosomal assembly/biogenesis (S6C Fig). Surprisingly, among the 85 down-regulated genes, most of those related to ribosomal assembly were found in previously published MYCN ChIP-seq data on mouse embryonic stem cells (ESCs) [33], strongly suggesting that these genes are direct targets of Mycn in zebrafish (Fig 5C). qPCR analysis further confirmed the down-

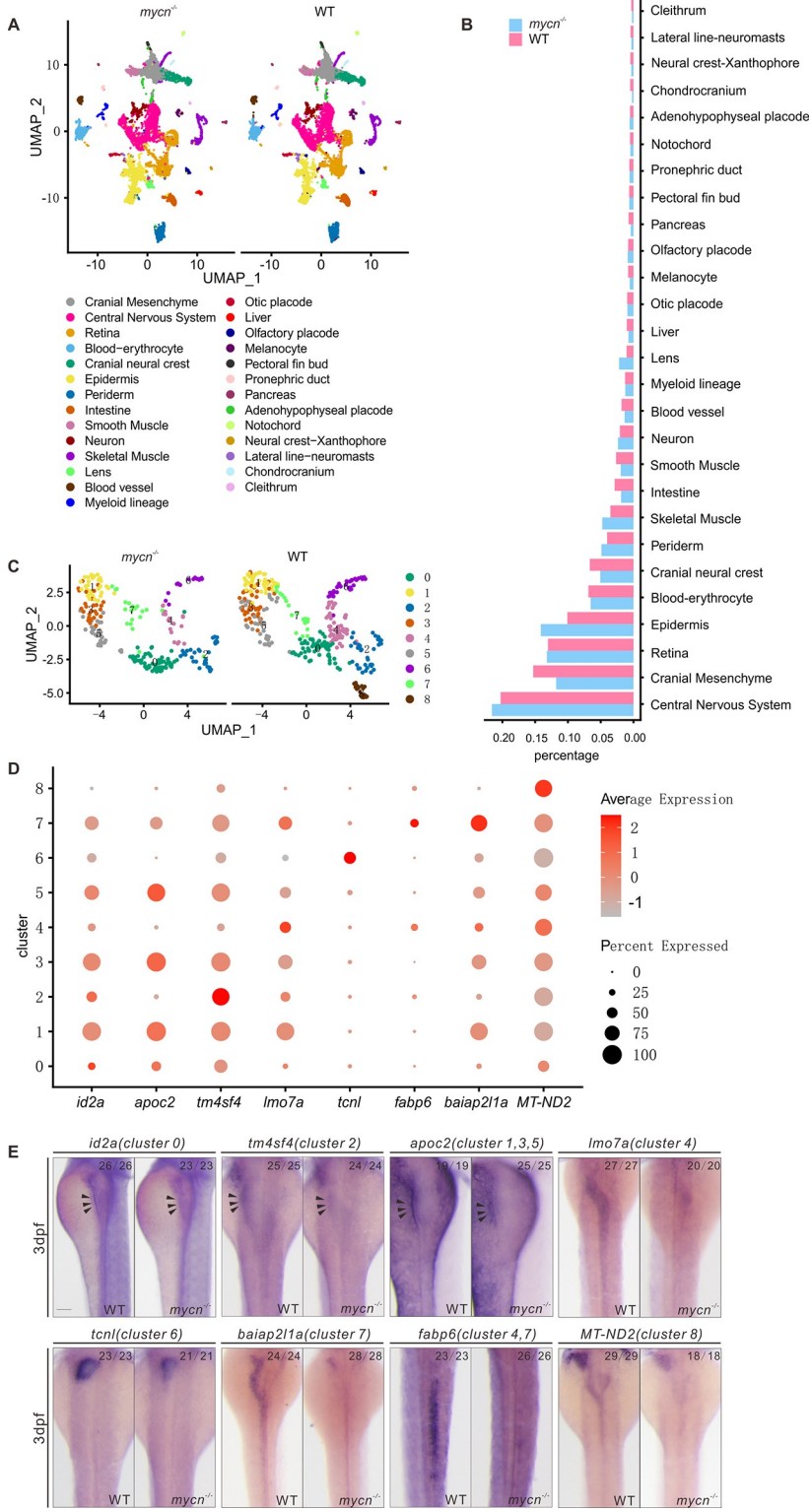

**Fig 3. Single-cell RNA-seq analysis of WT and *mycn* mutant embryos at 3 dpf. (A)** Unsupervised clustering of cells in the *mycn* mutants and WT embryos at 3 dpf. Cells are colored according to their cell type annotations inferred from expressed marker genes and published datasets. **(B)** Bar plot shows the percentages of each cell type in *mycn* mutants (blue) or WT embryos (red). **(C)** UMAP plot shows the subclusters of intestinal cells selected from A. Cells are colored by cell type clusters. A total of 9 clusters were identified by unsupervised clustering. **(D)** Dot plot shows the expressions

of marker genes in each subcluster of intestinal cells. Heatmap represents average expression level and dot size represents percentage of cell expression across *mycn* mutant and WT embryos. **(E)** WISH results show the expression of marker genes of each cluster: *id2a* (cluster 0), *tm4sf4* (cluster 2), *apoc2* (clusters 1, 3, and 5), *lmo7a* (cluster 4), *tcnl* (cluster 6), *baiap2l1a* (cluster 7), *fabp6* (clusters 4 and 7), and MT-ND2 (cluster 8) in WT and *mycn* mutant embryos. Arrow heads indicate the intestines. Scale bar: 100 μm. dpf, days postfertilization; UMAP, uniform manifold approximation and projection; WISH, whole mount *in situ* hybridization; WT, wild-type.

regulation of those ribosomal genes in the *mycn* mutants (Fig 5E), and, interestingly, ISH of those genes showed that they were all highly enriched in the developing intestines of the zebra-fish embryos (Fig 5F). Metabolomics analysis also showed abnormal metabolism of multiple amino acids in the *mycn* mutants (Fig 5G). In summary, these results strongly suggest that the intestinal phenotype in the *mycn* mutants may have been caused by impaired ribosomal machinery and protein translation.

## Impaired protein translation was the main cause of the intestinal defects in the *mycn* mutants

To validate whether ribosomal biogenesis and protein translation was impaired in the *mycn* mutants, we firstly investigated the rRNA processing in the WT and *mycn* mutant by northern blot using probes for 5′ETS, ITS1, and ITS2 regions of pre-rRNA, respectively (Fig 6A). We

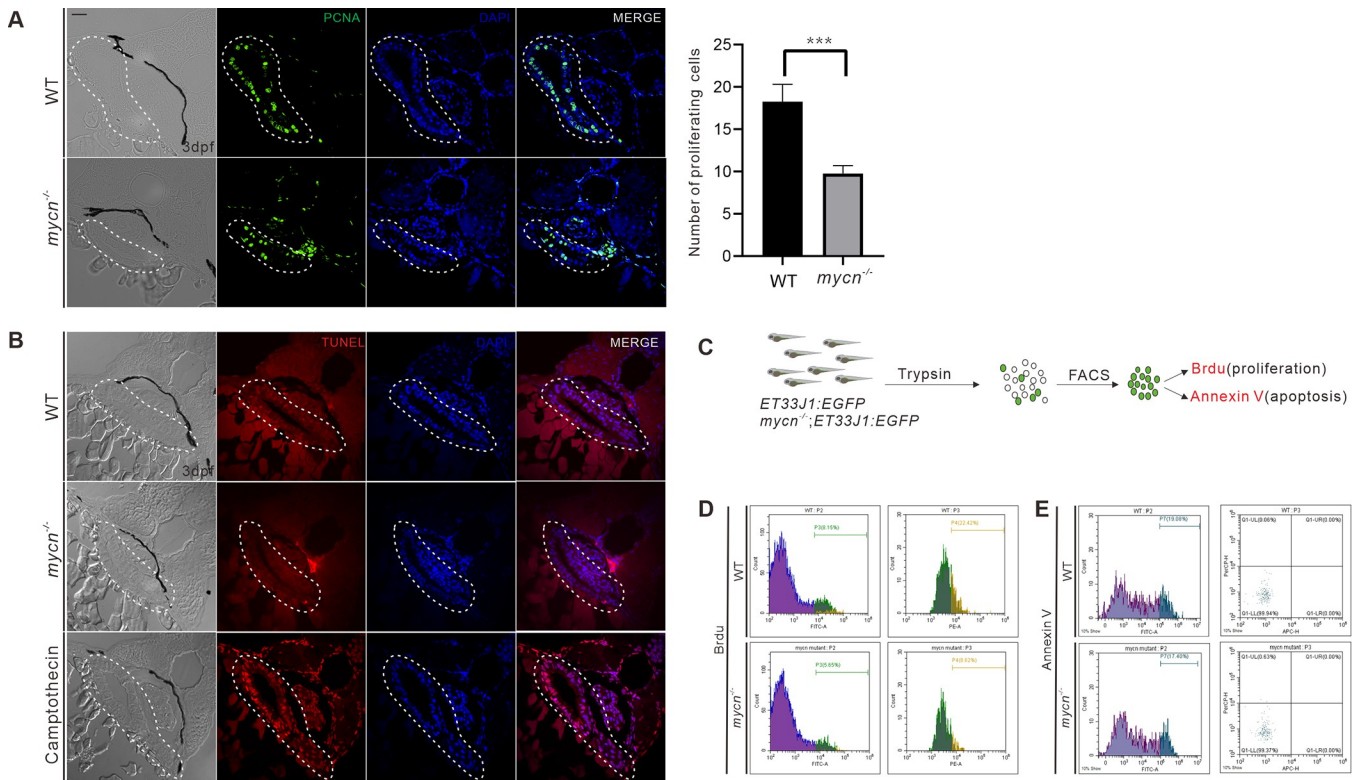

**Fig 4. Detection of cell proliferation and apoptosis in intestines of WT and *mycn* mutant. (A)** Cell proliferation in the intestines was detected by PCNA immunofluorescence staining (green signal) of the tissue sections from the WT and *mycn* mutant embryos. The statistical analysis of the proliferating cells of the sections of intestine in the right. **(B)** Apoptosis in the intestines was detected via TUNEL assay (red signal) for the tissue sections from the WT, *mycn* mutant, and camptothecin-treated (as positive control) embryos. Sections were cut along the transverse plane. Dotted lines indicate the intestine positions. **(C)** Schematic representation of the experimental workflow of the flow cytometry analysis. **(D)** Cell proliferation in the intestines was detected by BrdU incorporation assay. **(E)** Apoptosis in the intestines was detected by APC-Annexin V staining. Embryos used in the flow cytometry analysis were descendants of *ET33J1: EGFP* reporter line crossed with WT or *mycn* mutants (**D** and **E**). Scale bar: 50 μm. PCNA, proliferating cell nuclear antigen; WT, wild-type.

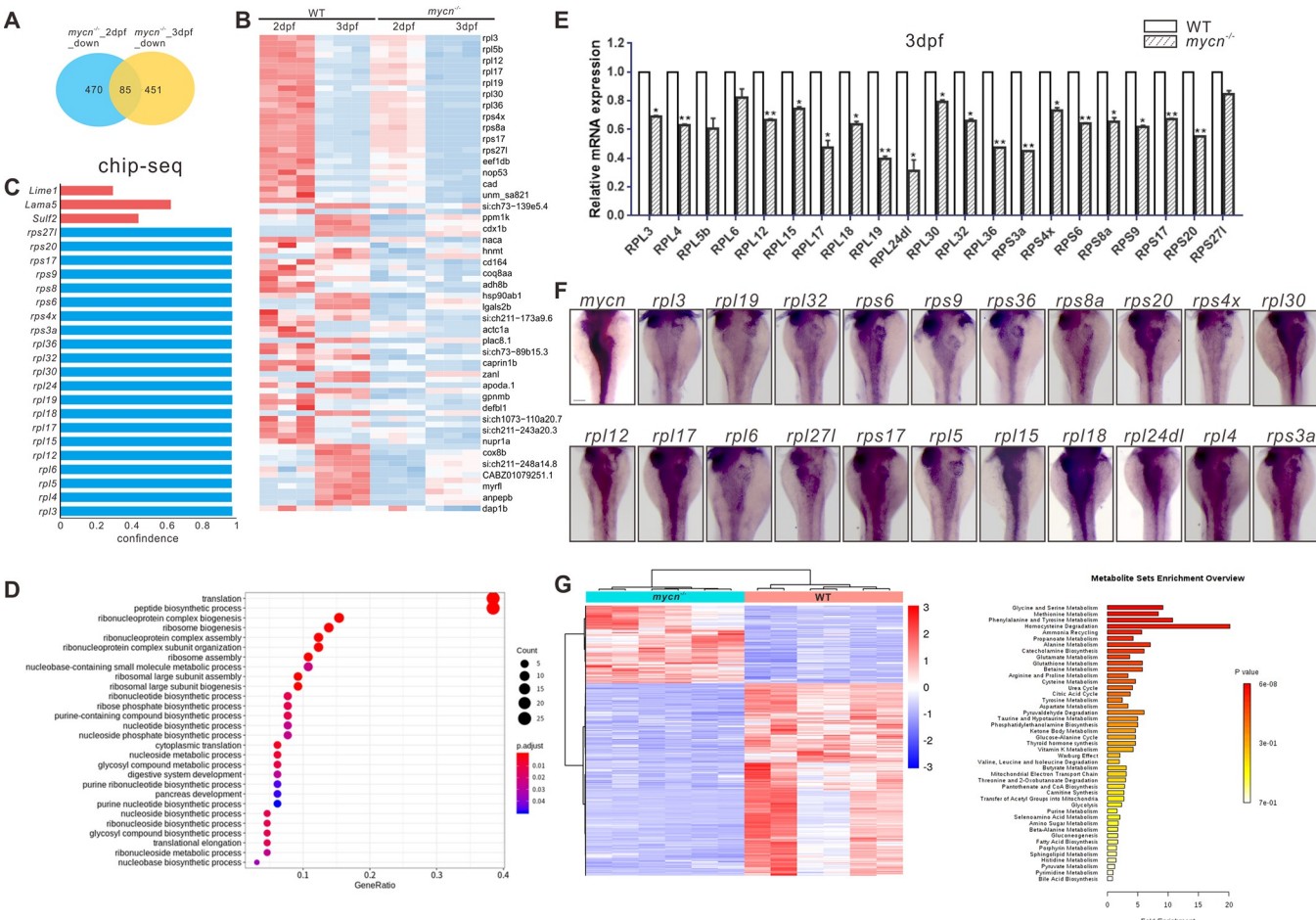

**Fig 5. Multiomics analysis in the *mycn* mutant and WT embryos. (A)** Venn diagram of the down-regulated genes in *mycn* mutants compared with WT at 2 and 3 dpf. The number of congruently down-regulated genes is shown in the middle. **(B)** Heatmap showing the scaled expression patterns of 85 down-regulated genes (identified in A) in *mycn* mutant and WT embryos at 2 and 3 dpf. Color scale: red, high expression; blue, low expression. **(C)** Bar plot shows the confidence interval of a portion of the *rpl* and *rps* genes that were direct downstream targets of MYCN analyzed by ChIP-seq of *MYCN* in the mouse ESC from published datasets [31]. These genes were also been found down-regulated in the *mycn* mutants by bulk RNA-seq analysis. **(D)** Dot plot showing enriched GO terms of the 85 down-regulated genes in the *mycn* mutants. The size and color intensity of each dot represents the gene counts in each enriched GO term and the adjusted *p*-value, respectively. **(E)** qPCR verification of the down-regulated *rpl* and *rps* genes at 3 dpf in WT and *mycn* mutant embryos. Asterisks indicate that the significant difference by Student *t* test. **$p < 0.001$; *$p < 0.05$. **(F)** Expression patterns of *rpl* and *rps* genes and *mycn* in the digestive organs of WT embryos at 3 dpf via WISH (images shown in dorsal view). **(G)** Metabolite analysis of the *mycn* mutant and WT embryos. The heatmap shows the expressions of different substances in *mycn* mutant and WT embryos after normalizing the intensity value of each metabolite. Bar plot shows the KEGG pathway enrichment and enrichment significance calculated by analysis of differential metabolites between the *mycn* mutant and WT embryos. Fold enrichment is the ratio of the number of metabolites matching the pathway during enrichment analysis to the number of theoretical metabolites distributed to the pathway of random distribution. Enrichment significance is indicated by color in the histogram according to *p*-value. Scale bar: 100 μm. The data underlying this figure can be found in S1 and S2 Data. dpf, days postfertilization; ESC, embryonic stem cell; GO, gene ontology; WISH, whole mount *in situ* hybridization; WT, wild-type.

did not see obvious aberrant intermediates of the processed pre-rRNA by ITS1 and ITS2 probe. Surprisingly, the 5′ETS-1 probe detected an extra smaller band, which is indicative of the impairment of rRNA processing. The impaired rRNA processing in *mycn* mutant may was a result of the decrease of rRNA processing–related genes (S6D and S6E Fig). Then, we conducted a sucrose density–gradient centrifugation assay to investigate the ribosomal profiling in both the mutant and WT embryos at 3 dpf. The amounts of free 40S, 60S, and 80S r-particles and polyribosomes were reduced in the *mycn* mutants (Fig 6B). This compromised ribosomal biogenesis further slowed down the synthesis of new proteins, as shown by western blotting of

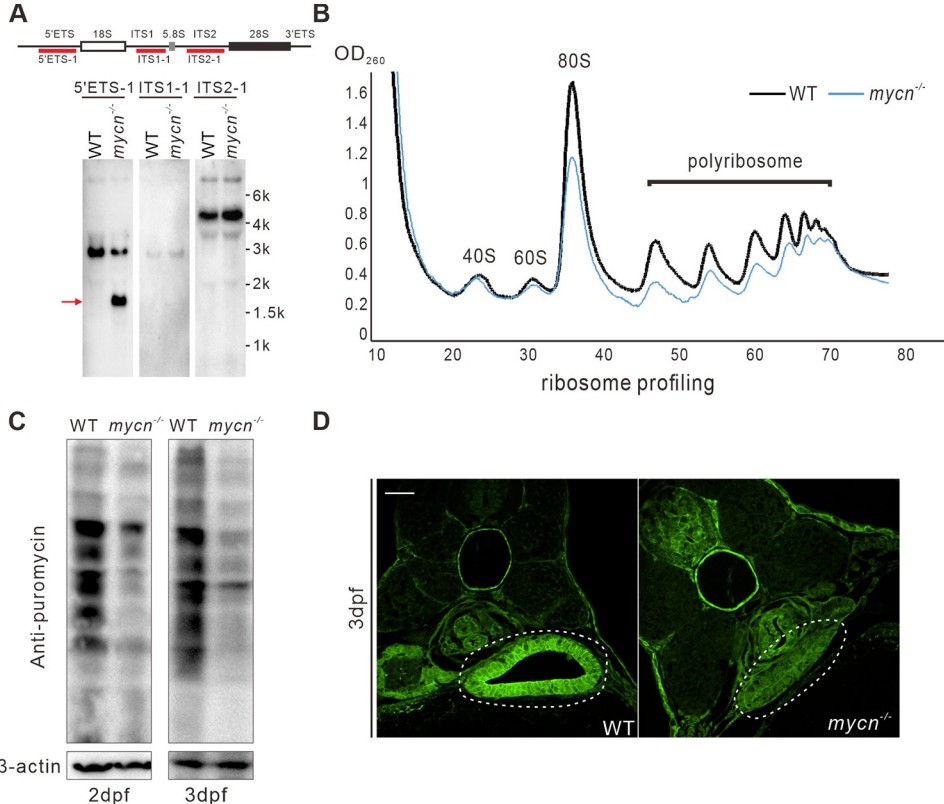

**Fig 6. Abnormal rRNA processing and impaired protein translation in *mycn* mutants. (A)** Northern blot analysis of *mycn* mutant and WT embryos at 5 dpf was performed using probe 5′ETS1, ITS1-1, and ITS2-1. Schematic drawing showing the structure of the pre-rRNA and probe positions of zebrafish (top) [34]. All samples were normalized by the total RNA. **(B)** Ribosomal profiling of *mycn* mutant and WT embryos at 3 dpf was performed by sucrose density–gradient centrifugation. All samples were normalized by the total RNA. **(C)** Detection of nascent protein synthesis by puromycin incorporation assay in WT and *mycn* mutant embryos at 2 and 3 dpf. Puromycin-incorporated neosynthesized proteins were detected by western blot with anti-puromycin antibody. The β-actin expression level was used as internal control. **(D)** Nascent protein synthesis was detected by section immunofluorescence with anti-puromycin antibody in WT and *mycn* mutant embryos at 3 dpf. Sections were cut along the transverse plane. All embryos are shown in dorsal view. Scale bar: 50 μm (**C**), 200 μm (**D-F**). Raw images of this figure are provided in S1 Raw Images. dpf, days postfertilization; WT, wild-type.

a puromycin incorporation assay (Fig 6C) and immunofluorescence by the anti-puromycin antibody of the intestinal sections (Fig 6D).

To further investigate the signaling pathways that have been affected by the impaired rRNA processing/ribosomal biogenesis in *mycn* mutant, we carried out Ribo-seq assay. For each gene, and for each sample, we calculated translation efficiency defined by the ratio of footprint FPKM to mRNA FPKM and further performed differential translation analysis between WT and *mycn* mutant embryos. We found that genes that had lower translation efficiency in *mycn* mutant embryos mainly enriched for signaling pathways including p53 and TGFbeta, which are important for cell death and proliferation [35,36], as well as the mTOR signaling pathway, which was reported to be involved in regulating ribosome biogenesis and translation (Figs 7A and 7B and S7B) [37]. Target of rapamycin (TOR) is a highly conserved serine/threonine kinase that regulates protein synthesis in response to various factors, including nutrients, growth factors, and amino acids [38,39]. mTORC1 controls protein synthesis by activating S6 kinase 1 (S6K1) and inhibiting 4E-binding protein 1 (4EBP1) [40]. Consistent with the Ribo-seq data, we found a slight decrease of rps6 protein but a dramatic decrease of phosphorylated

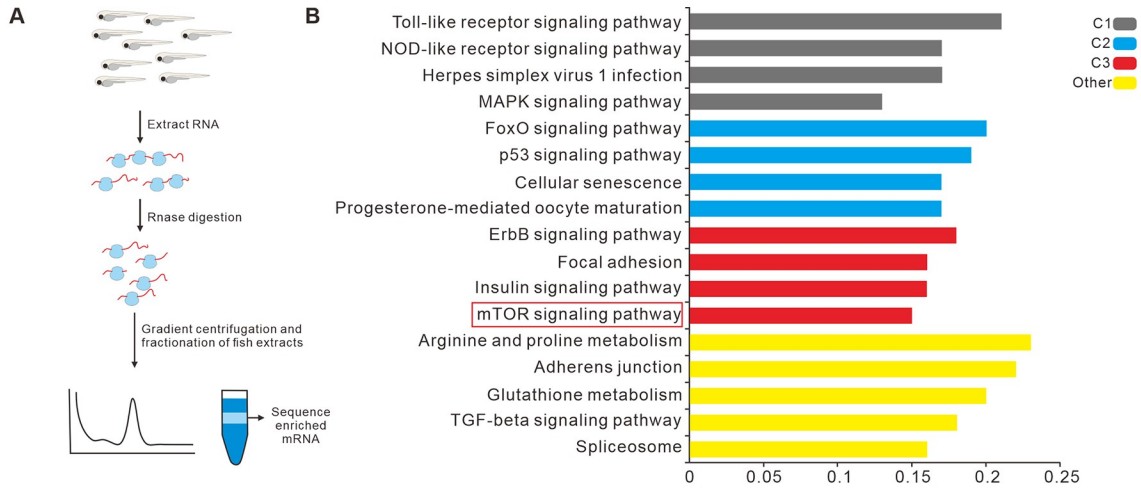

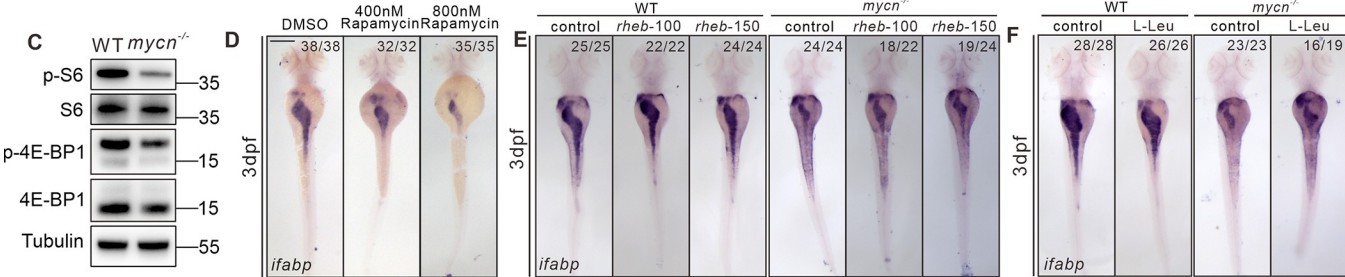

**Fig 7. Blockage of mTOR pathway and impaired protein translation results to intestinal defects in *mycn* mutants. (A)** Schematic representation of the experimental workflow of the Ribo-seq. **(B)** KEGG analysis of the genes showing reduced translation efficiency in the *mycn* mutant. The red box marks the mTOR pathway. **(C)** Detection of the rps6, phosphorylated rps6, eIF4EBP, and phosphorylated eIF4EBP in WT and *mycn* mutant embryos at 3 dpf. The proteins were detected by western blot. The tubulin expression level was used as internal control. **(D)** Inhibiting the mTOR pathway by different doses of rapamycin mimicked the intestinal defects phenotype of the *mycn* mutants. **(E, F)** Activating the mTOR pathway by injecting *rheb* mRNA and L-Leu treatment partially rescued the intestinal defects phenotype of the *mycn* mutants. All embryos are shown in dorsal view. Scale bar: 50 μm **(C)**, 200 μm **(D-F)**. Raw images of this figure are provided in S1 Raw Images. dpf, days postfertilization; *ifabp*, intestine fatty acid–binding protein; WT, wild-type.

rps6 and phosphorylated eIF4EBP in *mycn* mutant (Fig 7C), which further confirmed the impaired mTOR signaling pathway. We then found that treatment of rapamycin, the TOR inhibitor, could lead to intestinal developmental defects in zebrafish embryo, which was similar to *mycn* mutant, shown by ISH of *ifabp* (Fig 7D). Meanwhile, we found that some ribosome genes, which transcriptionally decreased in rapamycin-treated embryos, can be also found decreased in the *mycn* mutant (S7A Fig). These results suggest that proper protein synthesis is essential for intestinal development and that increased protein synthesis may rescue the intestinal defects in *mycn* mutants. Thus, we injected *rheb* mRNA and L-Leu, which can both forcefully activate the TOR pathway, into the *mycn* mutant embryos at 1-cell stage and 30 hpf, respectively. Surprisingly, both L-Leu and Rheb partially rescued the intestinal size in the *mycn* mutant embryos, as shown by *ifabp* expression (Fig 7E and 7F). However, we did not find an additive effect on the intestine defect by combining both supplements (S7D Fig). Although we did not find significant effect on the intestinal defect by injection of rps6 mRNA, we observed some minor rescue by injection of *rps3a* mRNA (S7C and S7E Fig). Besides, although using methotrexate to inhibit purine metabolism led to severe intestinal defects (S7F Fig), supplementing adenine and guanine in the *mycn* mutants did not restore the intestinal size (S7G Fig).

## Discussion

In this study, we generated *mycn* mutant fish mimicking the symptoms of human Feingold syndrome type 1. Among the many phenotypes caused by Mycn loss-of-function, we focused on intestinal atresia, a symptom that has the greatest impact on patients' quality of life yet has an unclear pathogenesis. Our model showed that Mycn plays a direct role in intestinal development by controlling the size of the entire digestive tract by regulating cell proliferation during development. The molecular mechanisms studying revealed that Mycn, as a transcriptional activator, directly regulates the transcription of genes related to ribosomal biosynthesis and assembly. Importantly, although inhibiting both protein translation and purine metabolism can lead to developmental defects in zebrafish intestines, the intestinal phenotype caused by Mycn loss-of-function was counteracted only when we used L-Leu or Rheb, which are activators of the TOR signaling pathway, to forcibly elevate the level of protein synthesis. This may provide a potential treatment strategy for alleviating intestinal defects in patients with Feingold syndrome.

Our work showed that during embryonic development, intestinal cells, which are in a highly proliferative state, require high *mycn* expression levels. Interestingly, *mycn* was also highly expressed in the neuroectoderm as early as gastrulation stage and continued to be enriched in the central nervous system during the organogenesis stage. However, we observed no obvious nervous system phenotypes in the *mycn* mutants, either morphologically or from single-cell sequencing. This might be because other MYC family members, such as c-myc or l-myc, which are also expressed in the nervous system, compensate for the function of Mycn. Future double or triple knockout of the Myc family members is required to validate this hypothesis and determine the relationship between the members of the MYC family. Another interesting question raised here is what regulates expression of the Myc genes, and what mechanism achieves the spatiotemporal expression of the Myc family members. Human Feingold syndrome is a dominant disease whereas a mutated *mycn* homozygous model of zebrafish can still show similar features. Our speculation is that, in zebrafish, the myc family members have multiple paralogues (*myca*, *mycb*, *mycla*, *myclb* and *mycn*) that may have redundant functions; this is probably the reason that the heterozygotes of *mycn* mutant do not show obvious phenotype.

In our *mycn*:*EGFP* reporter line, *mycn* was expressed in the migrating neuromasts of the lateral line, a sensory organ derived from neural crest cells in zebrafish, and our single-cell transcriptomic data showed that the cell ratios of multiple neural crest-derived organs were significantly lower in the *mycn* mutants. This is somewhat inconsistent with a previous study, which showed that MycN can drive the neural crest toward a neural stem cell–like fate in chicken embryos [6]. Based on our single-cell analysis of the *mycn* expression dynamics during development, we hypothesize that *mycn* expression levels might partially affect the fate of neural stem cells and the neural crest. When *mycn* expression is very high, the cells tend to differentiate into neural stem cells, and when the *mycn* expression is lower, the cells become a neural crest. But Mycn is essential for both cell types, and *mycn* deletion does not significantly affect the central nervous system because other Myc family members compensate for it.

Single-cell RNA-seq identified several clusters of developing intestinal cells that were highly sensitive to Mycn loss-of-function; these clusters were highly proliferating at this developmental stage and thus required a higher amount of *mycn*. Two recent works reported that mitochondrial transcription positively regulates intestinal development [41] and that copper overload affects intestinal development in zebrafish [42]. Among the intestinal clusters from our single-cell data, 1 cluster (cluster 8) expressing the highest level of *mycn* mRNA completely disappeared in the *mycn* mutants. Interestingly, many marker genes in this cluster are related

to mitochondrial function and copper metabolism (S2 Table), and this cluster may contain the cells that were mainly affected in the aforementioned 2 works. Our data provide a mechanistic understanding of these phenomena.

In summary, our work generated a zebrafish model of Feingold syndrome type 1. Our results suggest that proliferation arrest caused by protein synthesis blocking was the main reason for the developmental defects in the intestines of *mycn* mutant, suggesting a possible treatment strategy for intestinal symptoms in patients with Feingold syndrome. However, zebrafish were developed in vitro; thus, their nutrient acquisition and developmental environment differ from those of humans, and these results should be validated using a human intestinal organoid system.

## Supporting information

**S1 Fig. Expression pattern of *mycn* in zebrafish embryos. (A)** Expression of *mycn* shown by WISH in embryos at 6, 10, 18, and 24 hpf; AV, LV, and DV. **(B)** *mycn* mRNA expression levels in the zebrafish embryos were assessed via qPCR at different stages. **(C)** Schematic representing the EGFP knock-in strategy using CRISPR/Cas9 in zebrafish. **(D, E).** Analysis of expression dynamics of *mycn* during early development of zebrafish embryos using published single-cell RNA-seq datasets. *mycn* expression dynamics are shown in the URD tree (**D**) [27] and the zebrafish developmental landscape graph (**E**) [26]. **(F–I).** UMAP plot showing cell types of zebrafish embryonic scRNA-seq datasets at 2 dpf (**F**) and 5 dpf (**H**) [25]. Violin plots showing *mycn* expression levels in different cell types of WT embryonic scRNA-seq datasets at 2 dpf (**G**) and 5 dpf (**I**). Scale bars: 200 μm. AV, animal view; DV, dorsal view; hpf, hours postfertilization; LV, lateral view; WISH, whole mount *in situ* hybridization; WT, wild-type.
(TIF)

**S2 Fig. Construction of *mycn* mutant by CRISPR/Cas9 system. (A)** Schematic representation of the *mycn* gene editing strategy and the resulted mutation. Arrows indicate the PCR primers for mutation detection. **(B)** Verification of *mycn* mutants by PCR. The 505-bp strip represents the WT *mycn* fragment; the 197-bp strip represents mutated *mycn* fragment; and the lane of 2 strips (505 and 197 bp) represents the heterozygotes. **(C)** Survival curve of the WT and *mycn* mutants during the first 15 dpf development. **(D)** Representative photos of body length measurement of the WT and *mycn* mutant embryos at 72 hpf and 96 hpf. Statistics in the right panel. **(E)** Representative photos of eye area measurement of the WT and *mycn* mutant embryos at 96 hpf. Statistics in the right panel. **(F)** Alizarin red staining show the skull development in WT and *mycn* mutant embryos at 7 dpf, DV and LV. The data underlying this figure can be found in S1 Data. Raw images of this figure are provided in S1 Raw Images. dpf, days postfertilization; DV, dorsal view; hpf, hours postfertilization; LV, lateral view; WT, wild-type.
(TIF)

**S3 Fig. Morphological analysis of digestive organ by WISH in WT and *mycn* mutant. (A)** Expression of markers of digestive organs: *insulin* (islet), *prss1* (pancreas), and *lfabp* (liver) by WISH at 3 dpf in the *mycn* mutant and WT embryos. **(B)** Analysis of endoderm development by expression of pan-endoderm markers *foxa1*, *foxa3*, and *gata6* at 30 and 48 hpf. All embryos are in dorsal view. Scale bars: 200 μm. dpf, days postfertilization; hpf, hours postfertilization; WISH, whole mount *in situ* hybridization; WT, wild-type.
(TIF)

**S4 Fig. Phenotypic analysis in *mycn* mutants. (A)** Assessment of the effectiveness of protein and lipid digestion in the intestines of WT and *mycn* mutant embryos by EnzChek and PED6

treatment. The quenched fluorescent reporter PED6 or Enzchek is activated only after cleavage by intestinal phospholipase or protease. **(B, C).** Goblet cells (AB-PAS), absorptive cells (*slc15a1b*), and enteroendocrine cells (*pyyb*) in the intestines were analyzed by AB-PAS staining (left) and WISH (right) in WT and *mycn* mutant embryos at indicated developmental stages. Sections were cut along the sagittal plane. Arrow heads indicate the goblet cell. **(D)** Enteric neurons in WT and *mycn* mutant were shown by immunofluorescence with anti-Huc antibody at 4 and 5 dpf. All embryos are in lateral view. Scale bars: 200 μm (**A, C, D**), 50 μm (**B**). dpf, days postfertilization; WISH, whole mount *in situ* hybridization; WT, wild-type. (TIF)

**S5 Fig. scRNA-seq analysis for mycn mutant and WT embryos at 3 dpf.** (A) Dot plot showing expressions of 2 selected marker genes in each cell type. Dot size indicates the percentage of cells expressing the indicated genes; dot color indicates the average expression level of indicated genes. (B) Ridge plot showing the expression distributions of mycn in each subcluster of WT intestinal cells. (C) Pseudotime differentiation trajectory analysis for intestinal cells. (D) Spline plot representing changes in expression over pseudotime for the intestinal stem cell markers, sox9b and stat3. dpf, days postfertilization; scRNA-seq, single-cell RNA-seq; WT, wild-type. (TIF)

**S6 Fig. Bulk RNA-seq analysis in *mycn* mutants. (A)** Volcano plot showing DEGs between *mycn* mutant and WT embryos at 2 (left) and 3 dpf (right). DEGs were identified by |log2 (fold change) | > 0 and an adjusted $p < 0.1$. **(B)** Venn plot showing up-regulated genes in *mycn* mutants compared with WT at 2 and 3 dpf. The number of congruently up-regulated genes is indicated in the middle. **(C)** Emapplot showing interaction networks between enriched GO terms analyzed by 85 overlapped down-regulated genes in *mycn* mutants. The color scales indicate different thresholds of adjusted *p*-values, and the dot sizes represent the gene counts of each GO term. **(D, E)** GSEA of bulk RNA-seq datasets of mycn and WT at 48 hpf (**D**) and 72 hpf (**E**) using RNA processing signature. The data underlying this figure can be found in S2 Data. DEG, differentially expressed gene; dpf, days postfertilization; GO, gene ontology; GSEA, gene set enrichment analysis; hpf, hours postfertilization; WT, wild-type. (TIF)

**S7 Fig. Rescue experiments of intestinal development in *mycn* mutants. (A)** qPCR of *rpl* and *rps* genes at 3 dpf in WT and embryos treated with rapamycin. Asterisks indicate that the significant difference by Student *t* test. ***$p < 0.0001$; **$p < 0.001$; *$p < 0.05$. **(B)** Volcano plot shows genes that have differential translation efficiency between *mycn* mutant and WT embryos at 72 hpf. Blue dots and red dots indicate down-regulated genes and up-regulated genes in mycn mutants, respectively (*p*-value < 0.05, |log2 fold change| > 0.5). **(C)** Rescue experiment of the intestinal defects in the *mycn* mutants by injecting *rps6* mRNA. **(D)** Rescue experiment of the intestinal defects in the *mycn* mutants by activating the mTOR pathway: injecting *rheb* mRNA, or L-Leu and *rheb*/L-leu treatment. **(E)** Rescue experiment of the intestinal defects in the *mycn* mutants by injecting *rpl6*, *rpl24*, or *rps3a* mRNA. **(F)** Inhibition of purine de novo synthesis by methotrexate induces intestinal defects in zebrafish embryos. **(G)** Adenine (**A**) and guanine (**G**) supplementation cannot rescue the intestinal defects of *mycn* mutant. All embryos are in dorsal view. Scale bar: 200 μm. The data underlying this figure can be found in S1 and S2 Data. dpf, days postfertilization; hpf, hours postfertilization; WT, wild-type. (TIF)

**S1 Table. The primers used in this project including the probe, mRNA, knockout, and knock-in.**
(PDF)

**S2 Table. Marker genes of each cluster of intestine cells in WT and *mycn* mutant single-cell RNA-seq datasets.**
(PDF)

**S3 Table. Gene set differential expression analysis of *mycn* mutant versus WT samples using bulk RNA-seq expression data at 48 hpf.**
(PDF)

**S4 Table. Gene set differential expression analysis of *mycn* mutant versus WT samples using bulk RNA-seq expression data at 72 hpf.**
(PDF)

**S1 Data. The individual numerical values in Figs 2D and 5E and S2C–S2E and S7A.**
(XLSX)

**S2 Data. The individual numerical values in Figs 5A, 5B, and 5G and S6A and S7B.**
(XLSX)

**S1 Raw Images. Raw images of Figs 6A and 6C and 7C and S2B.**
(PDF)

## Acknowledgments

We thank Dr. Jin-Rong Peng, Dr. Jun Chen, Bao-Chun Su, and Dr. Ce Gao at Zhejiang University for providing the *ET33J1*: *EGFP* reporter fish, probe, and helpful discussion. We thank Jing-Yao Chen, Qiong Huang, and Xiao-Hui Chen from the Morphological Platform and Ying-Niang Li from the zebrafish core facility at Zhejiang University School of Medicine for their technical support. We thank Shao-Ge Luo at the Third Affiliated Hospital, Sun Yat-Sen University for his technical help. We thank Li-Yao Zhang and Qing-Hai Zhang at the Institute of Genetics and Department of Human Genetics, Zhejiang University School of Medicine for his technical help.

The content is the responsibility of the authors and does not necessarily represent the official views of any of the funding agencies.

## Author Contributions

**Data curation:** Yun-Fei Li, Tao Cheng, Ying-Jie Zhang, Xin-Xin Fu, Jing Mo, Guo-Qin Zhao, Mao-Guang Xue, Ding-Hao Zhuo, Yan-Yi Xing, Yang Dong, Xiao-Sheng Zhu.

**Formal analysis:** Yun-Fei Li, Tao Cheng.

**Funding acquisition:** Peng-Fei Xu.

**Investigation:** Yun-Fei Li, Tao Cheng, Ying-Jie Zhang, Xin-Xin Fu, Ying Huang, Xiao-Zhi Sun, Dan Wang, Xiang Liu, Dong Chen.

**Methodology:** Yun-Fei Li, Mao-Guang Xue, Feng He, Jun Ma.

**Resources:** Jun Ma.

**Software:** Tao Cheng.

**Supervision:** Xi Jin, Peng-Fei Xu.

**Writing – original draft:** Yun-Fei Li, Peng-Fei Xu.

**Writing – review & editing:** Yun-Fei Li, Tao Cheng, Xi Jin, Peng-Fei Xu.

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
