## [Editor Report · Decision Letter 0]

18 May 2022

Dear Pengfei, 

Thank you for submitting your manuscript entitled "Mycn regulates intestinal development through ribosomal biogenesis in a zebrafish model of Feingold syndrome 1" for consideration as a Research Article by PLOS Biology.

Your manuscript, reviews from Review Commons, and revision plan have now been evaluated by the PLOS Biology editorial staff as well as by an Academic Editor with relevant expertise and I am writing to let you know that we would like to consider a revised version of your manuscript that addresses the reviewer comments from Review Commons.

Before we can invite you to submit a revised manuscript, we need you to complete your submission by providing the metadata that is required for full assessment. To this end, please login to Editorial Manager where you will find the paper in the 'Submissions Needing Revisions' folder on your homepage. Please click 'Revise Submission' from the Action Links and complete all additional questions in the submission questionnaire.

Once you have completed your submission, we will send you a formal "major revision" decision, and a 3 month deadline for the revision.

Please re-submit your manuscript within two working days, i.e. by May 20 2022 11:59PM.

Kind regards,

Luke

Lucas Smith

Associate Editor

PLOS Biology

lsmith@plos.org

---

## [Editor Report · Decision Letter 1]

23 May 2022

Dear Pengfei,

Thank you for submitting your manuscript "Mycn regulates intestinal development through ribosomal biogenesis in a zebrafish model of Feingold syndrome 1" for consideration as a Research Article at PLOS Biology. As mentioned in our last email, your manuscript, the reviews from Review Commons, and your revision plan have been evaluated by the PLOS Biology editors and an Academic Editor with relevant expertise.

In light of the reviews, which I have appended to the end of this email, we would like to invite you to revise the work to thoroughly address the reviewers' reports, as outlined in your submission.

Given the extent of revision needed, we cannot make a decision about publication until we have seen the revised manuscript and your response to the reviewers' comments. Your revised manuscript is likely to be sent for further evaluation by all or a subset of the reviewers.

**IMPORTANT - SUBMITTING YOUR REVISION**

*Re-submission Checklist*

*Published Peer Review*

*PLOS Data Policy*

*Blot and Gel Data Policy*

Sincerely,

Lucas

Lucas Smith

Associate Editor

PLOS Biology

lsmith@plos.org

REVIEWS:

Reviewer 1: 

--Evidence, reproducibility and clarity--

**Major**

What percent of mice survive to adulthood, what is the cause of death in those that do not survive? In fertile survivors, do progeny all survive to adulthood or is there again a significant number that die prior to adulthood. What is the basis for the heterogeneity in survival?

Any insights into mTOR signaling, relative effects on S6 vs eIF4E? I think that authors did polysome profiling, not ribosome profiling, as few specific targets are discussed. If authors have knowledge of specific targets, this would improve the manuscript.

--Significance--

Careful analysis of developmental trajectory and effects of mycn knockout in fish, with single cell RNAseq data, and some analysis of translation control and the proteome.

The analysis of GI development is new, much of the effects on ribosome biogenesis is known in mammals, but not in fish, and rescues are new.

Mycn enthusiasts, medical geneticsts and fish people

I am a physician scientist interested in MYCN and cancer

Reviewer 2: 

--Evidence, reproducibility and clarity--

The manuscript of Li Y-F and al.; describes the role of ribosomal biogenesis in a zebrafish model of Feingold syndrome. The work is quite appreciable for the production of a new animal model to investigate this syndrome, but the conclusions seem not appropriate. The main molecular defect displayed by this model is represented by a decrease in ribosomal gene expression that the author associate with a decrease in ribosome biogenesis. However, the authors do not show any experiments about rRNA processing. Has this model a defect in 90S ribosomal formation? Moreover, establishing a direct role of ribosome biogenesis should improve ribosome biogenesis to rescue the nonfunctional myc model. By rapamycin, the authors decrease the protein synthesis to mimic similar defects observed in the zebrafish, they did not show if a decrease of ribosome gene expression occurs. Morever, they are aging on protein synthesis and do not on ribosome biogenesis.

--Significance--

The manuscript proposes a new vertebrate model to investigate the Feingold syndrome, but at molecular level is poor investigated

Reviewer 3: 

--Evidence, reproducibility and clarity--

**Summary**

This study presents a loss-of-function model in zebrafish for Feingold syndrome. The phenotype is described in detail (in situ hybridization, reporter lines and histology) and by using different -omics techniques (bulk and single-cell transcriptomics, metabolomics and genomics). An impressive amount of validation (mainly using in situ hybridization was performed. Moreover, the authors uncover clues towards the pathophysiological mechanism. In addition, a putative treatment is also suggested.

**Major comments**

1. Quantification/statistical evaluation.

There is a consistent lack of quantification in this study, which is very problematic. Many of the phenotypic claims in the model are illustrated in a qualitative way. This is unacceptable. Especially since quantification is possible in most cases. For example:

Figure 1A Eyes are smaller at 4dpf; please quantify and statistically evaluate if this is a significant difference

Figure 1C Quantify length of intestine (GFP signal) and statistically evaluate. Figure 4 Claims are made regarding proliferation (PCNA) and apoptosis (TUNEL). Although the figures look convincing, there is NO quantification presented although this can easily be done. The authors should provide quantifications and statistical evaluation of this data before claiming that there are phenotypic differences. The same holds true for the flow cytometry data in Figure 3D

Figure 6 Quantification and appropriate statistical evaluation is lacking again. The data in Figure 6A an B, D and F should be quantified. How can one assess reproducibility and proportion of a change if no quantification, nor statistics is provided?

It is also unclear how many biological and technical replicates of an experiment have been performed. Please provide this information.

2. More focus on skeletal abnormalities. Are there any abnormalities in the fins or axial skeleton. Especially the fins need to be studied as the hands are highly affected in Feingold syndrome. Short stature is also a common feature; why not measure length in zebrafish larvae to verify this? Why not quantify Fig1B (craniofacial skeleton)? Measuring the angles of different craniofacial elements is a common way of quantification when assessing abnormalities in the craniofacioal structures of zebrafish larvae.

3. How could it be explained that Feingold syndrome is a dominant disease whereas the model in zebrafish is a homozygous (recessive) model and still show similar features?

4. How can it be ruled out that the morphological features (short intestine, no swim bladder, smaller eyes) observed in knockout embryos are not due to a global developmental delay in the knockouts?

5. It is an important result (if supported by quantification) to see that supplementation with Leucine or Rheb-150 shown improvement of the phenotype. Does this supplementation also improve the skeletal features in zebrafish? Is there an additive/synergistic effect if one would combine both supplements?

**Minor comments**

1. In figure 2, a dramatic reduction of ifabp is described. Could this reduction been found back in the transcriptome data?

2. Results section; second paragraph: "some of the homozygotic mycn mutant fish..." Homozygotic should read homozygous.

--Significance--

Description of relevant disease model for Feingold syndrome. The authors use state- of-the-art techniques (single cell-sequencing) and established genomic techniques (Tol2 trans genesis, Crispr/Cas9, reporter lines, in situ hybridization).

The data presented looks convincing and seems of good quality. However, because of a lack of quantifications it is impossible to determine if the results presented are representative for the vast majority of mutant vs normal embryos.

This study provides some important insights in the pathophysiology. In addition it provides a hint towards experimental therapy.

Expertise:

Human clinical genetics/ human disease modeling in zebrafish/ zebrafish genomics/ metabolic disease/ heart and neurological disease

---

## [Editor Report · Decision Letter 2]

8 Sep 2022

Dear Dr Xu,

Thank you for your patience while we considered your revised manuscript "Mycn regulates intestinal development through ribosomal biogenesis in a zebrafish model of Feingold syndrome 1" for publication as a Research Article at PLOS Biology. This revised version of your manuscript has been evaluated by the PLOS Biology editors and the Academic Editor. We are pleased to say that the Academic Editor is satisfied by the changes made in the revision and feels the manuscript is now ready for publication. 

**However, before we can accept your manuscript, we need you to address the following editorial requests in a revision that we think will not take very long. Please attend to the following editorial requests: 

1) Financial Disclosures: In your financial disclosure statement, please describe the role of any sponsors or funders in the study design, data collection and analysis, decision to publish, or preparation of the manuscript. If the funders had no role in any of the above, include this sentence at the end of your statement: "The funders had no role in study design, data collection and analysis, decision to publish, or preparation of the manuscript."

2) Ethics statement: In your ethics statement, please provide the protocol number that was approved by the Animal Ethics Committee of the School of Medicine, Zhejiang University.

3) Data request: You may be aware of the PLOS Data Policy, which requires that all data be made available without restriction: http://journals.plos.org/plosbiology/s/data-availability. For more information, please also see this editorial: http://dx.doi.org/10.1371/journal.pbio.1001797

a) Supplementary files (e.g., excel). Please ensure that all data files are uploaded as 'Supporting Information' and are invariably referred to (in the manuscript, figure legends, and the Description field when uploading your files) using the following format verbatim: S1 Data, S2 Data, etc. Multiple panels of a single or even several figures can be included as multiple sheets in one excel file that is saved using exactly the following convention: S1_Data.xlsx (using an underscore).

b) Deposition in a publicly available repository. Please also provide the accession code or a reviewer link so that we may view your data before publication. 

Fig 2D; Fig 5A,E,G; Fig S2C-E; Fig S6A; Fig S7A-B

>>Please also ensure that figure legends in your manuscript include information on where the underlying data can be found, and ensure your supplemental data file/s has a legend.

>>Please ensure that your Data Statement in the submission system accurately describes where your data can be found.

4) Data request: We require the original, uncropped and minimally adjusted images supporting all blot and gel results reported in an article's figures or Supporting Information files. We will require these files before a manuscript can be accepted so please prepare and upload them now. Please carefully read our guidelines for how to prepare and upload this data: https://journals.plos.org/plosbiology/s/figures#loc-blot-and-gel-reporting-requirements

>>Please provide the uncropped images corresponding to Fig 6A,C; Fig 7C; Fig S2B

We expect to receive your revised manuscript within two weeks. 

*Published Peer Review History*

*Press*

Sincerely,

Lucas

Lucas Smith, Ph.D.

Associate Editor,

lsmith@plos.org,

PLOS Biology

---

## [Editor Report · Decision Letter 3]

27 Sep 2022

Dear Dr Xu,

Thank you for the submission of your revised Research Article "Mycn regulates intestinal development through ribosomal biogenesis in a zebrafish model of Feingold syndrome 1" for publication in PLOS Biology. On behalf of my colleagues and the Academic Editor, Bon-Kyoung Koo, I am pleased to say that we can in principle accept your manuscript for publication, provided you address any remaining formatting and reporting issues. These will be detailed in an email you should receive within 2-3 business days from our colleagues in the journal operations team; no action is required from you until then. Please note that we will not be able to formally accept your manuscript and schedule it for publication until you have completed any requested changes.

**IMPORTANT: As you address the formatting and reporting requests, to come, we also ask that you address the following request: 

1) Thank you for providing the underlying data for your figures in supplementary files S1_data and S2_data. Can you please reference these in the relevant figure legends (including supplemental)? For example, to each figure legend, you could add the sentence "the data underlying this figure can be found in supplementary files S1_data and S2_data".

PRESS

Sincerely, 

Lucas Smith, Ph.D., Ph.D.

Associate Editor

PLOS Biology

lsmith@plos.org